# Reducing target binding affinity improves the therapeutic index of anti-MET antibody-drug conjugate in tumor bearing animals

Amita Datta-Mannan[1]*, Hiuwan Choi[2], Zhaoyan Jin[3], Ling Liu[4], Jirong Lu[4], David J. Stokell[4], Anthony T. Murphy[3], Kenneth W. Dunn[5], Michelle M. Martinez[5], Yiqing Feng[4]

1 Exploratory Medicine and Pharmacology, Lilly Corporate Center, Indianapolis, IN, United States of America, 2 Bioproduct Research & Development, Lilly Technology Center North, Indianapolis, IN, United States of America, 3 Drug Disposition/Commercialization, Lilly Corporate Center, Indianapolis, IN, United States of America, 4 Biotechnology Discovery Research, Lilly Technology Center North, Indianapolis, IN, United States of America, 5 Department of Medicine, Division of Nephrology, Indiana University School of Medicine, Indianapolis, IN, United States of America

* datta_amita@lilly.com

**Data Availability Statement:** All relevant data are within the paper and its Supporting Information files.

## Abstract

Many oncology antibody-drug conjugates (ADCs) have failed to demonstrate efficacy in clinic because of dose-limiting toxicity caused by uptake into healthy tissues. We developed an approach that harnesses ADC affinity to broaden the therapeutic index (TI) using two anti-mesenchymal-epithelial transition factor (MET) monoclonal antibodies (mAbs) with high affinity (HAV) or low affinity (LAV) conjugated to monomethyl auristatin E (MMAE). The estimated TI for LAV-ADC was at least 3 times greater than the HAV-ADC. The LAV- and HAV-ADCs showed similar levels of anti-tumor activity in the xenograft model, while the $^{111}$In-DTPA studies showed similar amounts of the ADCs in HT29 tumors. Although the LAV-ADC has ~2-fold slower blood clearance than the HAV-ADC, higher liver toxicity was observed with HAV-ADC. While the SPECT/CT $^{111}$In- and $^{124}$I- DTPA findings showed HAV-ADC has higher accumulation and rapid clearance in normal tissues, intravital microscopy (IVM) studies confirmed HAV mAb accumulates within hepatic sinusoidal endothelial cells while the LAV mAb does not. These results demonstrated that lowering the MET binding affinity provides a larger TI for MET-ADC. Decreasing the affinity of the ADC reduces the target mediated drug disposition (TMDD) to MET expressed in normal tissues while maintaining uptake/delivery to the tumor. This approach can be applied to multiple ADCs to improve the clinical outcomes.

## Introduction

Antibody-drug conjugates (ADCs) are a promising class of therapeutics that coalesce the antigen binding specificity of monoclonal antibodies (mAbs) with the potency of cytotoxic small molecules [1, 2]. With >10 approved ADC drugs (reviewed in [3] and many additional ADCs in clinical studies, these medicines are becoming increasingly important therapeutic modalities

 

**Funding:** The author(s) received no specific funding for this work.

**Competing interests:** The authors have declared that no competing interests exist.

for treatment of various malignant tumors and hematological malignancies [4–6]. The success of ADC drugs depends on establishing a reasonable therapeutic window through optimizing the delicate balance of the selective delivery of highly toxic chemotherapeutics (or payload) to cancer cells, while sparing normal cells from the risk of side effects. Antibody-drug conjugate toxicity can result from premature release of the cytotoxic agent in circulation, as well as, ADC uptake and payload release into healthy cells. As a means to mitigate the former, several chemical and engineering strategies have been applied successfully to optimize conjugation chemistries and ADC peripheral linker stability, thereby substantially reducing premature systemic payload release [7]. With regard to the distribution of ADCs into healthy cells, the antibody component plays a major mechanistic role and thus, is largely affected by factors influencing the disposition of mAbs, including a mixture of non-target related fluid phase endocytosis, non-specific interactions (i.e. charge- and hydrophobic-based interactions with cells), binding to Fcγ receptors based on the antibody isotype choice, interaction with the neonatal Fc receptor (FcRn), and target mediated drug disposition (TMDD). While an ideal target for an ADC is highly expressed exclusively on tumor cells, validated targets have some level of expression in normal tissues [8, 9]. Hence, while various mAb engineering and screening strategies can be leveraged to reduce target independent normal tissue uptake, TMDD facilitates uptake into the tissue expressing the antigen even when expressed at a low level. Significant partitioning and uptake of ADCs into normal tissue likely decreases the therapeutic window through both reducing efficacy and increasing toxicity [10, 11].

Dysregulation of the mesenchymal-epithelial transition factor or MET (also known as hepatocyte growth factor [HGF] receptor) signaling pathway has been implicated in numerous studies as a key contributor in cancer biology including proliferation, metastasis and resistance to therapy in non-small cell lung cancers (NSCLC), colorectal cancers (CRC), and head and neck cancers (HNSCC) [12–15]. While exceptionally promising from a mechanistic perspective, MET directed mAb monotherapies have not been successful due to limited tumor regression efficacy [16, 17]. However, the broad spectrum of MET-expressing tumors makes MET an attractive target for ADC modalities to deliver potent cytotoxic agents. Since MET is also widely expressed in healthy tissues including liver and kidney [8], it raises significant concerns around the viability of an ADC therapeutic directed at MET to achieve a reasonable therapeutic window. Indeed, a number of recent publications [18–22] have shown the promise of targeting MET to deliver cytotoxic agents to tumor. Findings from the clinical studies of ABBV-399, an ADC consisting of the anti–MET antibody ABT-700 and monomethyl auristatin E (MMAE), have demonstrated, as a monotherapy [18], safety and tolerability profiles and promising antitumor activity in patients, which resulted in breakthrough therapy designation by the FDA in 2022. However, based on the dose limiting toxicities observed at 3.0 and 3.3 mg/kg the recommended phase 2 dose for ABBV-399 was established at 1.9 mg/kg every 2 weeks and 2.7 mg/kg every 3 weeks, indicating a narrow therapeutic window.

A potential broadly applicable approach for improving the therapeutic window for ADCs targeting antigens that are endogenously expressed in normal tissues, such as MET, involves decreasing specific ADC uptake in non-tumor tissue by reducing the mAb binding affinity to the target. Lowering the affinity of mAbs often slows the peripheral clearance rate and subsequent biodistribution through reduced TMDD within the tissues [23]. In the case of ADCs, lowering the target binding affinity of the mAb component may also mitigate the on-target, off-tumor toxicity due to decreased TMDD within normal tissues, while maintaining robust pharmacological efficacy through sufficient TMDD to tumor tissue which often over-expresses the antigen for adequate delivery of the potent cytotoxic agent to tumors. It has been demonstrated for tomoregulin (TENB2), a promising molecular target for prostate cancer that is also expressed in healthy gastrointestinal tissue (GIT), that pre-administration of the unconjugated

TENB2 mAb, at a suitable dose level, saturates the low-to-moderate peripheral antigen expression in the GIT while maintaining tumor uptake and efficacy of the TEN2B-ADC in antigen overexpressing rodent explant models [11]. A recent publication using trastuzumab-based ADCs to test the benefit of coadministration of trastuzumab in preclinical models concluded that coadministration of the naked antibody improved the ADC efficacy for tumor with high antigen expression levels [24]. While there is no published non-clinical safety data or clinical feasibility information around the effect of the pre-administration or coadministration approach on the ADC's therapeutic window, these findings are encouraging and support the concept that ADC efficacy can be retained while reducing the normal tissue uptake of ADCs.

In the present work, we studied the effect of mAb affinity to MET in MET-ADCs for both on-target toxicity and antitumor activity in rodent models. Because MET is a target antigen that is expressed in many tumor types but also found in healthy tissues, leveraging MET targeting ADCs serves as a platform for interrogating the role that the target binding affinity of the antibody component of ADCs can have on the therapeutic window of these modalities (Fig 1). We demonstrate that (i) there is less accumulation of the low-affinity variant (LAV) relative to the high-affinity variant (HAV) MET mAb in liver by IVM using fluorescently-tagged mAbs, (ii) the low-affinity MET-ADC (LAV-ADC) has tumor regression activity comparable to a high-affinity MET-ADC (HAV-ADC), (iii) the LAV-ADC has improved safety signals relative to the HAV-ADC, (iv) the comparable efficacy and improved safety of the LAV-ADC imparts the LAV-ADC molecule with a minimally ~3-times wider therapeutic index than the HAV-ADC counterpart and (v) the underlying mechanistic basis of the safety and efficacy findings were illuminated via radiolabeled ADC in normal tissues and tumor by quantitative and single photon emission computed tomography (SPECT) uptake and elimination evaluations, as well as free payload concentration measurements by mass spectroscopy. These particularly promising findings revealed a potential clinically applicable strategy to increase the ADC therapeutic window using existing linker-payload technologies and warrant additional interrogations to determine the appropriate target affinity for ADC therapies that will reduce ADC distribution and toxicological liabilities in antigen-expressing healthy tissues.

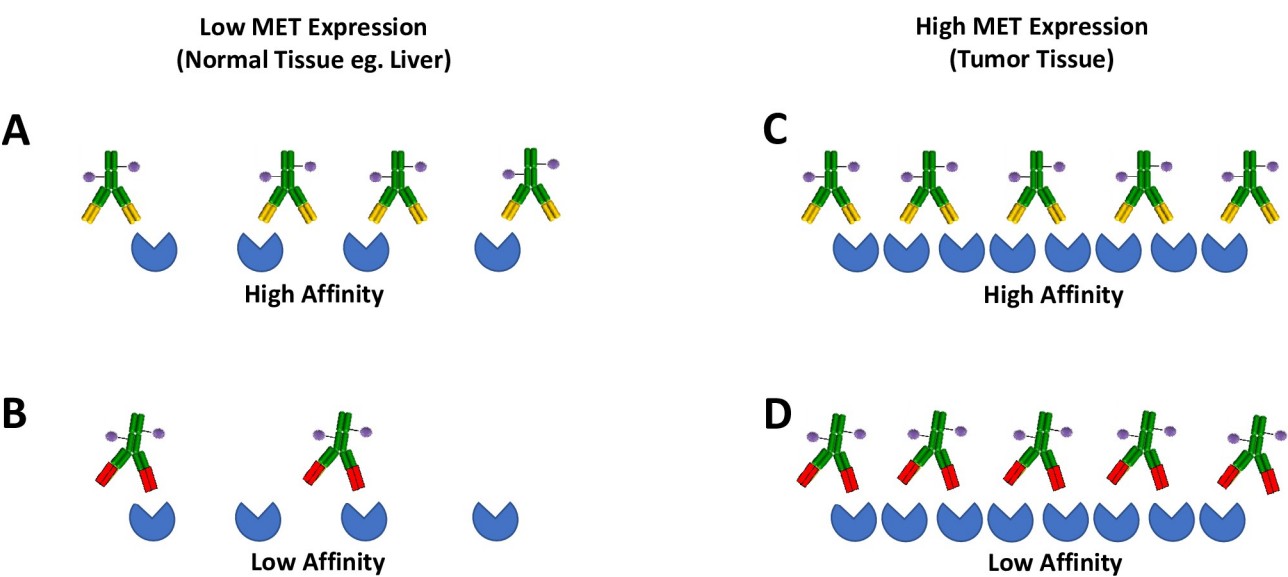

**Fig 1. Schematic depiction of the potential influence of target binding affinity on the therapeutic window of an ADC.** Decreasing the affinity of the mAb component of the ADC can reduce the specific binding to MET expressed in normal tissue or TMDD (A and B) while maintaining sufficient uptake of the ADC to tumor which over-expresses MET likely due to avidity (C and D).

## Materials and methods

### Antibody and conjugate preparation

The high affinity and low affinity mAbs against MET, henceforth known as HAV and LAV, respectively, and an isotype control antibody were expressed in Chinese hamster ovary cells, purified using standard purification procedure, and conjugated to MMAE as described in Datta-Mannan et al. [25]. Each ADC sample was buffer-exchanged into phosphate buffer saline (PBS) after conjugation and analyzed by analytical size exclusion chromatography (SEC), analytical hydrophobic interaction chromatography (HIC), and liquid chromatography mass spectrometry (LC-MS). The aggregation level for all mAb and ADC samples were <5% by analytical SEC. The average DAR numbers of the MET-ADCs were determined to be 3.2–3.6 and the isotype control ADC 3.8–4.3 by HIC and LC-MS.

HAV and LAV mAbs used for IVM studies were labeled with DyLight488 (DL488) NHS Ester (Thermo Fisher, Waltham, MA) using the previously reported protocol, followed by preparative size exclusion chromatography [26], resulting in average dye to mAb ratio of 0.9 and 1.0, respectively. The aggregation levels of the fluorescent-tagged mAbs were <1% by analytical SEC.

### Binding affinity of HAV and LAV, DL488 fluorescently labeled HAV and LAV, and HAV- and LAV-ADCs to human and rat MET

Binding affinity of the mAbs, fluorescent-tagged mAbs, and ADCs to human or rat MET extracellular domain (ECD) were determined using surface plasmon resonance (SPR) on a Biacore T200 instrument (Biacore Life Sciences, Pittsburgh, PA, USA). All MET-ECD were made at Eli Lilly and Company (proprietary reagents). Except as noted, all reagents and materials were from GE Healthcare Bio-sciences Corp. Biacore experiments were performed at 37°C in HBS-EP+ buffer (GE Healthcare, 10 mM HEPES, pH 7.4, 150 mM NaCl, 0.005% PS20, and 3 mM EDTA). A CM5 sensor chip containing immobilized protein A, generated using standard NHS-EDC amine coupling on all four flow cells (Fc), was employed as capture methodology. The mAb, fluorescent-tagged mAb, or ADC was captured by Protein A chip to yield about 100 response units. Multiple cycles of different concentrations of MET-ECD (6.25–200 nM) were then injected over flow cells. Each cycle includes a mAb, fluorescent-tagged mAb, or ADC capture step followed by the injection of MET-ECD at a single concentration with a 20-minute dissociation period, followed by a regeneration step using glycine HCl (pH 1.5). Bulk refractive index differences were corrected for by subtracting the response obtained in a reference flow cell, where no anti-MET antibody was captured. Data were fit to a 1:1 binding model using Biacore T200 Evaluation software version3.1 to determine the association rate ($k_{on}$, $M^{-1}$ $s^{-1}$ units), dissociation rate ($k_{off}$, $s^{-1}$ units), and $R_{max}$ (RU units). The equilibrium dissociation constant ($K_D$) was calculated from the relationship $K_D = k_{off} / k_{on}$.

### IVM studies of cellular distribution in rats

IVM studies used a total of 38 Munich-Wistar-Fromter rats (9–12 weeks of age) from a colony maintained at Indiana University that was originally derived from animals generously provided by Dr. Roland Blantz (UC San Diego). All animals were maintained at the Indiana University LARC facility and were provided with food and water *ad libitum*. All animal experiments were approved and conducted according to the Institutional Animal Care and Use Committee guidelines of Indiana University and adhered to the guide for the care and use of animals standards set by the National Institutes of Health [27]. Animals which displayed signs of failure to thrive (greater than 20% weight loss, refusal to eat, lack of spontaneous

activity, labored breathing not relieved with appropriate analgesic, vocalization not relieved with appropriate analgesic, or lack of socialization) were euthanized. No animals died before meeting the euthanasia criteria. In housing, animal health and behavior were monitored daily, and animal weight measured every other day. During IVM studies, animal health and behavior were monitored every 10 minutes.

IVM studies were conducted either 24 hours after intravenous (IV) injection of DL488 labeled HAV or LAV (6 mg/kg), respectively, or during and for the first two hours following IV injection. Studies were conducted generally as described previously [28, 29]. Surgical preparation was started approximately one hour prior to imaging. For studies of the liver, the rat was placed on an induction chamber connected to an anesthesia isoflurane circuit at 2–4% of isoflurane with 300–600 ml/min $O_2$. Once stabilized under anesthesia, the ventral abdominal side of the body and the right side of the neck were shaved and cleaned. Prior to exposing the liver, a jugular catheter was prepared for probe injections. A 1 cm right ventral incision was made in the neck, the jugular was exposed and all fat and fascia surrounding were cleared. The anterior end of the jugular was tied using 4–0 silk suture to prevent bleeding. A tiny nick was made in the jugular vein and a catheter was slid roughly 1 cm into the jugular vein and secured at the posterior end using 4–0 suture. The catheter was sutured and secured to the skin in three different places. After the jugular cannula was placed, the liver was exposed for imaging. A 2x2 cm piece of gauze moistened with saline was glued with a drop of cyanoacrylate to the skin just below the sternum. Exposure of the liver was begun with a 1 cm lateral ventral incision below the rib and the skin and muscle layers were removed. The liver was then exposed by gently squeezing through the incision and placed on the moistened gauze. The liver was placed in a 40 mm cover-glass bottomed dish (WillCo Well) and glued to the cover glass without pressure. The rat was then transferred to the microscope stage and placed on warming pads under a heat lamp to maintain body temperature at 35–37° Celsius, as monitored using a rectal probe thermometer. The microscope objective was also maintained at 37° Celsius via an objective heater.

For studies of rat kidney, a jugular catheter was placed as described above. Following jugular catheterization, a 1 cm small lateral incision was made over the left kidney, followed by a 0.5 cm incision in the peritoneal muscle. The kidney was identified and pulled up through the skin opening. Once the kidney was exposed, the adrenal gland and the renal ligament was separated. The kidney was then placed in a 40 mm cover-glass bottomed dish with a 2x2 cm piece of gauze moistened with saline. For many of the studies shown here, the same rat was imaged at both 0–2 and again 24 hours after injection. For these studies, following imaging at 2 hours, the rat kidney was replaced into the body, the incision sutured, and the rat returned to housing until the next day.

IVM imaging studies were conducted using a Leica TCS SP8 DIVE confocal/ multiphoton system mounted on an inverted stand. The imaging was performed using a 25X NA 0.95 water immersion or 63X NA1.3 glycerol immersion Leica HCX APO objective lenses. Images were collected using multiphoton fluorescence excitation at 920 nm. Three channels of fluorescence were collected in non-de-scanned detectors using emission bandpass filters of 405–450 nm (blue), 500–550 nm (green) and 600–650 nm (red). Image volumes were collected using 600 Hz bidirectional scanning, at different zoom factors and a vertical spacing of 1.5 microns. For studies of the early stages of antibody disposition, a series of images (typically 465x465 microns in size) were collected during injection and 3x3 image volume mosaics (spanning ~1.4 by 1.4 mm to a depth of 50 microns) were collected prior to, and then again 1 hour and 2 hours following IV injection. For studies conducted 24 hours after IV injection several 3x3 image volume mosaics were collected. For IVM studies, animals were anesthetized using isoflurane (3–4% for induction, 1–3% for maintenance). The depth of anesthesia was verified by toe pinch

and adjusted as necessary. At the end of the IVM studies, animals were euthanized by isoflurane overdose followed by cervical dislocation. At the end of imaging for each timepoint, rats were euthanized. No animals died prior to euthanization. The rats were perfuse-fixed and liver and/or kidney tissue was collected.

## Rat HT-29 xenograft model

Rat HT-29 xenograft studies were designed with a total of 60 animals (of which 48 were used on study) and were conducted at Covance (Greenfield, IN; now Labcorp Inc.) and were designed and executed within accordance of the Animal Use Protocol (AUP) and adherence to the Covance Institutional Animal Care and Use Committee (IACUC) regulations and abided by the standards set by the National Institutes of Health [27]. Animals were provided with food and water *ad libitum*. Animals which displayed signs of failure to thrive (greater than 20% weight loss, refusal to eat, lack of spontaneous activity, labored breathing not relieved with appropriate analgesic, vocalization not relieved with appropriate analgesic, or lack of socialization) were euthanized. No animals died before meeting the euthanasia criteria. In housing and on days of tumor volume collection, animal health and behavior were monitored daily and animal weight measured every other day. HT-29 cells were purchased for research purposes from the American Type Culture Connection (ATCC). HT-29 is a human colon cancer cell line used extensively in biological and cancer research and is known to express MET [8]. The HT-29 derived xenografts were established through subcutaneous inoculation of 5 x $10^6$ cells suspension in a 1:1 cell suspension: matrix ratio with Matrigel Matrix Basement Membrane. Dosing was initiated when mean tumor volumes reached 250–350 mm$^3$ in NIH-Nude rats (Taconic).

## HT-29 xenograft rat tumor regression study and tumor immunofluorescence staining

The aforementioned HT-29 xenograft rat model (with a total of 60 animals) was used to evaluate the tumor growth inhibition of the HAV and LAV MET-ADCs, which were administered as single agents. Three dose levels of each ADC were evaluated: 3-, 6- and 10-mg/kg (same doses as for biodistribution, pharmacokinetic and toxicology studies). When tumors reached a volume of 250–350 mm$^3$, 40 animals were randomized into groups of 6 rats each (8 rats in the 10 mg/kg groups only) and given an IV bolus injection of the test materials. An additional group of 8 rats (from the total of 48 animals on study) received 10 mg/kg of an isotype control ADC. Tumor volume growth was monitored for up to 24 days post-dose. Animals that did not have the intended tumor volumes (12 animals) were euthanized by isoflurane overdose followed by cervical dislocation prior to randomization and administration of the test materials.

Tumor length (l, the longest dimension) and width (w, perpendicular to the length) were measured by calipers; tumor volume V was approximated as $V = 0.536*l*w^2$. Sparse PK samples of plasma were collected on days 1, 2, and 7 post ADC dose and analyzed for the concentrations of total antibody or conjugated, as previously described [30]. Rats were euthanized before tumors became ulcerated or reached the maximum allowable volume (1000 mm$^3$). At the end of the study or at the time of tumor collection, animals were euthanized by isoflurane overdose followed by cervical dislocation. No animals died before meeting the euthanasia criteria.

Tumor biopsy samples were collected from 2 animals each in the 10 mg/kg isotype control ADC, the HAV and LAV MET-ADC groups at 24 hours post-dose. Tumor samples from the animals were transferred to 10% NBF (neutral buffered formalin solution) for up to 72 hours at ambient conditions for fixation. The tumor tissues were then embedded in paraffin. Following the embedding procedure, the paraffin blocks were sectioned at a thickness of 5 μm and placed

onto slides using standard procedures. Prior to staining, the formalin fixed paraffin embedded (FFPE) slides were processed and antigen retrieval was conducted. A Dako Autostainer (Carpinteria, CA) was programmed to stain all slides for the detection of human IgG or MET in the tumor tissue samples. Stained slides were coverslipped using Dako fluorescent mounting media. Images of the stained slides were collected on a 3-D HISTECH (Budapest, Hungary) scanner having Plan-Aplchromat 40x objective lenses at ambient temperature. The fluorochromes used were Alexa-488, Alexa-647 and DAPI. A PCO.edge camera was used to capture images in the JPEG medium and the Pannoramic Viewer software (3DHISTECH) was used for image acquisition. Brightness and contrast parameters were applied consistently to all images.

## Normal Sprague Dawley rat pharmacokinetic and toxicology studies

Normal Sprague Dawley rats (total of 18) were obtained from The Jackson Laboratory (Bar Harbor, ME). All rats were treatment-naive male between the ages of 8 to 11 weeks with an average weight of 0.3 kg (+/- 0.05 kg). Studies were conducted at Covance (Madison, WI; now Labcorp Inc.) and adherence to the Covance Institutional Animal Care and Use Committee (IACUC) regulations and abided by the standards set by the National Institutes of Health [27]. Animals were provided with food and water *ad libitum* throughout. Animals that displayed signs of failure to thrive (greater than 20% weight loss, refusal to eat, lack of spontaneous activity, labored breathing not relieved with appropriate analgesic, vocalization not relieved with appropriate analgesic, or lack of socialization) were planned to be euthanized; however no animals displayed these signs. No animals died before meeting the euthanasia criteria. In housing, animal health and behavior were monitored daily and animal weight measured every other day.

The HAV and LAV MET-ADCs were dosed IV at 3-, 6- and 10-mg/kg with a dose volume of 1 mL/kg (dose prepared in PBS pH 7.4). Blood samples for pharmacokinetic evaluations for each ADC were collected using serial sampling from the jugular vein at 0.083, 6, 24, 72, 96, 120, 168, 216 and 336 hours after dose administration from three rats/timepoint under an isoflurane anesthesia (3–4% for induction, 1–3% for maintenance). The depth of anesthesia was verified by toe pinch and adjusted as necessary. Animals were stabilized under isoflurane overdose and euthanized by cervical dislocation after collection of the 336-hour terminal post administration samples. The pharmacokinetic blood samples were collected into tubes containing sodium EDTA as anticoagulant and processed to plasma for subsequent concentration analyses. Blood samples for clinical chemistry evaluations, such as alanine transaminase (ALT) and aspartate transaminase (AST), were collected from the jugular vein prior to administration of the ADCs and at 24, 72, 168 and 336 hours following dose administration from three rats/timepoint and processed into appropriate tubes for clinical chemistry evaluations.

Based on best practices set by an industry white paper [30], two assays were developed to describe the plasma PK for the ADCs, including total human IgG (Tab) and the IgG conjugated payload. The total IgG concentrations and IgG conjugated payload were determined using a bead-based anti-human IgG immuno-affinity capture. Briefly, a biotinylated goat anti-human IgG (Fc specific) (Southern Biotech) was used to capture the ADCs. Complexes were then removed from the plasma using streptavidin coated magnetic beads. The beads were washed and the ADCs were eluted from the anti-human IgG capture antibody. The total IgG samples were then reduced, alkylated and treated with trypsin prior to detection of the ADC heavy chain using a Q-Exactive Plus Orbitrap mass spectrometer (Thermo). For the IgG conjugated drug assay, following immuno-affinity capture, samples were treated with Cathepsin B which leads to the proteolysis of the Cathepsin B sensitive component of the linker and liberates the MMAE payload from the antibody. Following the cleavage of the linker, the free payload was quantitated by LC-MS/MS to determine the IgG conjugated payload concentrations.

Pharmacokinetic parameters for each of the ADCs were calculated using the WinNonlin Professional (Version 3.2) software package (Pharsight Corporation, Mountain View, CA). Plasma concentration-time data were calculated using a model-independent approach based on the statistical moment theory. The parameters calculated included the maximum plasma concentration ($C_{max}$), area under the curve ($AUC_{inf}$), clearance (CL), volume of distribution at steady state ($V_{ss}$) and elimination half-life ($t_{1/2}$).

## $^{124}$I- and $^{111}$In-DTPA-radiolabeled LAV- and HAV-ADC HT29 xenograft rat biodistribution and SPECT/CT studies

The HT29 xenograft rat model was also used to evaluate the tissue distribution of the HAV and LAV MET-ADCs following a single administration of each compound at a dose level of 6-mg/kg (a total of 30 animals were used). The study was conducted in accordance with SOPs and the protocol as approved by Eli Lilly and Company and adherence to the Covance Institutional Animal Care and Use Committee (IACUC) regulations and abided by the standards set by the National Institutes of Health [27]. Animals were provided with food and water *ad libitum* throughout. Animals that displayed signs of failure to thrive (greater than 20% weight loss, refusal to eat, lack of spontaneous activity, labored breathing not relieved with appropriate analgesic, vocalization not relieved with appropriate analgesic, or lack of socialization) were planned to be euthanized; however no animals displayed these signs. No animals died before meeting the euthanasia criteria. In housing, animal health and behavior were monitored daily and animal weight measured every other day. The studies were in compliance with the requirements contained in the MPI Research (now Charles River Laboratories) Radioactive Materials License Number 21-11315-02, and all applicable regulations issued by the Nuclear Regulatory Commission (NRC). Briefly, the non-radiolabeled HAV and LAV MET-ADCs were conjugated to diethylene triamine pentaacetic acid (DTPA) and radiolabeled with $^{111}$In at MPI Research, Inc. to target a low specific activity. To prepare the dosing formulations, the appropriate amount of radiolabeled test article was combined with the required volume of vehicle, phosphate buffered saline (PBS) and mixed. The dosing formulations were administered once via IV injection into the tail vein of HT-29 tumor bearing rats. Doses were labeled to a target dose levels of ~10 μCi/animal (6 mg/kg).

Blood samples (~200 to 300 μL) (cohorts of three animals per timepoint per group) were collected from all animals using cardiac puncture after carbon dioxide inhalation (terminal samples for tissue collections at each timepoint) into tubes containing sodium citrate. Prior to maxillary vein blood collection and cardiac puncture euthanasia, rats were stabilized under an isoflurane anesthesia overdose. All samples were analyzed using the gamma counter to determine residual radioactivity. Blood samples were collected at 0.083, 24, 72, 96 and 168 hours post-dose. The total weight of each blood sample was recorded and analyzed for radioactivity using the gamma counter. The %ID/g values were corrected for radioactive decay over time. The liver, kidney and tumor were also collected from the three animals per time point per group (0.083, 24, 72, 96 and 168 hours post-dose), weighed and analyzed for total radioactivity via the gamma counter. Similar to the blood samples, the %ID/g values for the tissues were corrected for radioactive decay over time.

A subset of the animals were analyzed by SPECT/CT scans prior to tissue collection. Individual gamma and dose calibrator radioactivity counts were utilized for image reconstruction and analysis. Animals were anesthetized with 2lpm Oxygen/1.5 to 2% isoflurane and eye lube was placed on the animals prior to scanning. Static SPECT scans were performed on three animals per timepoint per group at 0.083, 24, 72, 96 and 168 hours post-dose for 60 minutes. Each static scan was followed by a CT scan for anatomical reference. After each scan time point,

each animal was placed in the dose calibrator and the total remaining reactivity was recorded. SPECT/CT data were transferred to inviCRO, LLC for image analysis.

### Free payload exposure in liver and tumor tissues of LAV- or HAV-ADC-dosed HT29 xenograft rats

Frozen liver and tumor tissues (from three animals per timepoint per group) from the aforementioned HT29 xenograft rat biodistribution study were pulverized on dry ice and then weighed. The pulvzerized tissues were further homogenized in PBS containing a protease inhibitor cocktail. Mixture of acetonitrile and methanol saturated with ascorbic acid were added to extract free payload. The extraction was concentrated under nitrogen and re-suspended in 20% formic acid prior to injection to LC-MS/MS to measure free MMAE.

## Results

### Design and *in vitro* characterization of MET antibodies and MET-ADCs with varying target affinity

For the studies herein, two MET mAbs with varying target binding affinity, originating from mouse immunization, were humanized into human IgG2 antibodies, and subsequently fluorescently labeled via amine coupling or conjugated to the potent antimitotic agent MMAE via interchain cysteine residues. The low- and high-affinity anti-MET antibodies, designated as LAV and HAV respectively, differ at a single amino acid residue in the complementary domain regions (Eli Lilly proprietary sequences). The binding affinities of the mAbs, fluorescently labeled mAbs, and ADCs to the MET (ECD) were measured by surface plasmon resonance (Table 1). HAV binds to human and rat MET with $K_D$ values of approximately 4 and 3 nM, respectively, while LAV binds with reduced affinities to human and rat MET ($K_D$ values of approximately 165 and 121 nM, respectively). The binding affinity of the mAbs upon MMAE conjugation or fluorescent labeling were comparable to their naked antibodies (Table 1). The reasonably similar ~40-fold difference in human and rat MET $K_D$ values between the HAV and LAV as well as between the corresponding ADCs facilitated using these molecules for non-clinical studies of their therapeutic windows in human xenograft tumor bearing rat models.

### IVM studies of the cellular disposition of DL488-labeled HAV and LAV in rat liver and kidney

MET is widely expressed in healthy tissues including liver and kidney [8]. As such, in order to identify the cells and subcellular compartments in which LAV and HAV accumulate as a

**Table 1. Mean rat and human MET binding affinities for mAbs and ADCs.**

| Molecule | Binding Affinity $K_D$ (nM) | |
|---|---|---|
| | **Rat MET** | **Human MET** |
| **HAV** | 3.1 | 4 |
| **LAV** | 121 | 165 |
| **DL488-HAV** | 3.1 | ND |
| **DL488-LAV** | 112 | ND |
| **HAV ADC** | 2.2 | 4 |
| **LAV ADC** | 77 | 165 |

ND = not determined.

surrogate of their ADC counterparts' disposition, IVM studies in rats were conducted following IV injection of DL488-labeled HAV and LAV. Fig 2 shows examples of multiphoton fluorescence images collected from the livers of living Munich Wistar Fromter rats 24 hours after IV injection with DL488-HAV (panels A and C) or DL488-LAV (panels B and D). These images clearly demonstrate that DL488-HAV accumulates in the hepatic sinusoids, appearing as bright puncta in the higher-magnification image. DL488-LAV likewise accumulates in sinusoidal puncta, but to a modest degree in only a few segments.

In order to determine whether the constructs interact transiently with the liver, IVM of the liver was conducted over the first two hours following IV injection. Consistent with the results shown in Fig 2, these studies demonstrate that DL488-HAV, but not DL488-LAV accumulates in endothelial puncta that are detectable within the first hour following injection (S1 Fig). Images shown in both Figs 2 and S1 illustrate a diffuse fluorescence of DL488-LAV in the sinusoid lumens consistent with LAV being predominantly within the peripheral blood circulation. At no time is either construct detectable in hepatocytes (identified by their brown autofluorescence).

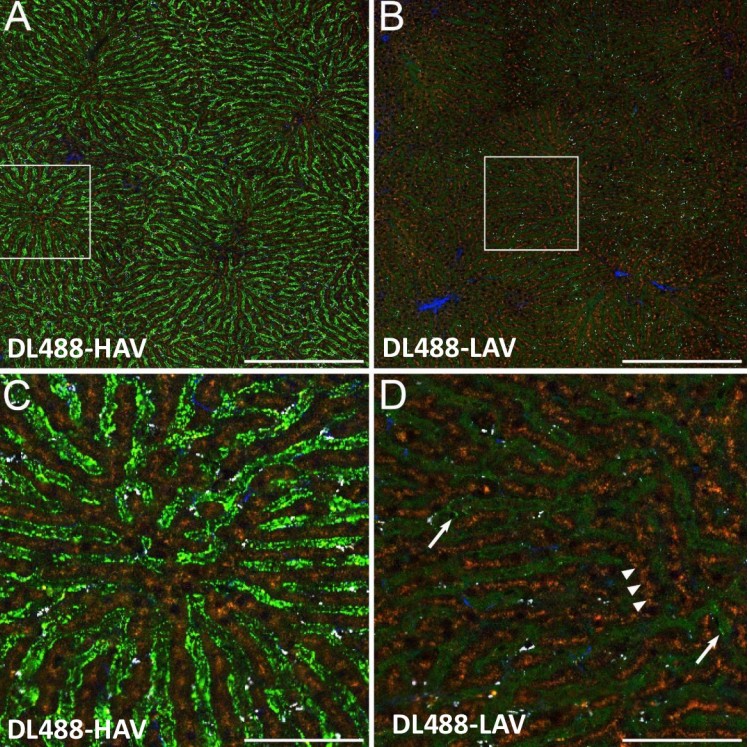

**Fig 2. Distribution of fluorescently-labeled HAV and LAV in the liver of living rats.** Multiphoton fluorescence excitation images of the livers of living Munich-Wistar-Fromter rats were collected 24 hours after intravenous injection of DL488-HAV or DL488-LAV (6 mg/kg). DL488-HAV accumulates in punctate (bright green) structures of sinusoidal endothelia (Panel A, but more apparent in the 4X magnified region shown in panel C). In contrast, DL488-LAV is observed primarily free in the lumens of the sinusoids (Panels B and D), with only minor amounts of punctate endothelial accumulation (indicated with arrows in the magnified region shown in panel D). Neither construct is detectably associated with hepatocytes, which can be identified by their brown autofluorescence, arranged in linear "cords" (arrowheads in panel D). Bright white punctate fluorescence reflects broad-spectrum autofluorescence of vitamin A in stellate cells. Blue signal derives from second harmonic generation, likely from collagen. Scale bars represent 400 microns in panels A and B, and 100 microns in panels C and D. All images are collected using identical microscope settings and contrast enhanced identically.

IVM studies were next conducted to compare the disposition of DL488-HAV and DL488-LAV in the liver with that in the kidney. Images were collected from the kidney and then the liver of the same rats 24 hours after injection with either DL488-HAV or DL488-LAV (Fig 3). Consistent with the results shown in Fig 2, both probes accumulated in puncta in the sinusoidal endothelia of the liver, but whereas DL488-LAV accumulated to a modest degree in only a few segments (Fig 3B), DL488-HAV accumulated to high levels in endothelia throughout the liver (Fig 3A). Images of the kidneys of these same rats are shown in Fig 3C and 3D. The rodent kidney is noteworthy for the bright yellow-orange autofluorescence stimulated by multi-photon excitation, arising from lysosomes of proximal tubule epithelia. The endogenous nature of this characteristic fluorescence is demonstrated in Fig 3, which shows that it is present prior to probe injection, and unaltered following injection. Using this autofluorescence as a landmark one can readily identify the lumens and epithelial cells of proximal tubules, the network of peri-tubular capillaries (occupying the space between proximal tubule epithelia), and distal tubules (which are essentially non-fluorescent). These images demonstrate that, in contrast to the liver, neither construct accumulates detectibly in the endothelia of the peritubular capillaries. Both probes are conspicuously absent from tubule lumens with the exception of a few distal tubule segments (indicated with arrows), indicating minimal renal filtration. As observed in the liver, images of the kidney show a diffuse fluorescence of DL488-LAV in the lumens of the peritubu-lar capillaries, consistent with LAV disposition predominantly within the blood circulation.

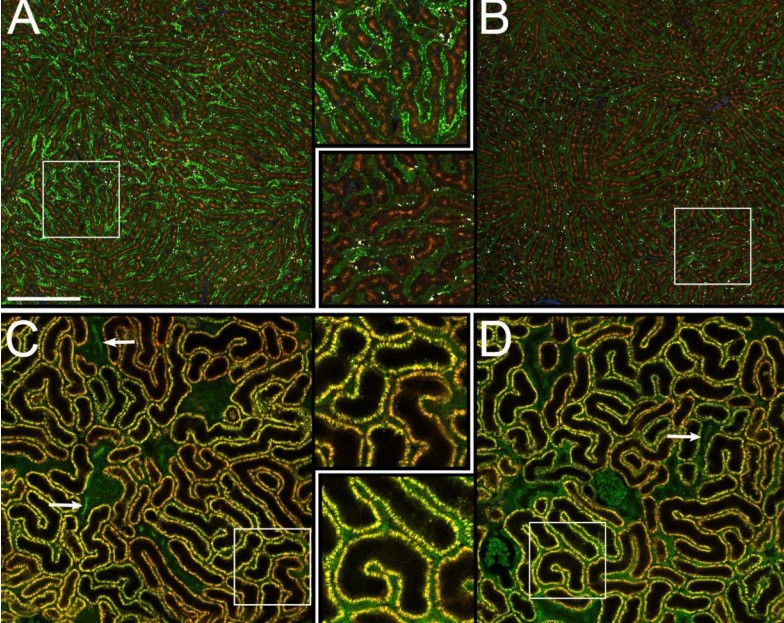

**Fig 3. Distribution of DL488 fluorescently-labeled HAV and LAV in the liver and kidney of the same living rats.** Multiphoton fluorescence excitation images of the kidneys of living Munich-Wistar-Fromter rats were collected 24 hours after intravenous injection of 6 mg/kg DL488-HAV or DL488-LAV. The livers of these same rats were then presented for intravital microscopy of the liver. (A) DL488-HAV in the liver. (B) DL488-LAV in the liver. (C) DL488-HAV in the kidney of the same rat as shown in panel A. (D) DL488-LAV in the kidney of the same rat as shown in panel B. Indicated regions are reproduced at 2x magnification in the center column. Brown fluorescence in the liver results from hepatocyte autofluorescence, and yellow-orange fluorescence in the kidney results from proximal tubule autofluorescence. Arrows indicate distal tubules (identified as segments lacking autofluorescence) with low levels of probe fluorescence, indicating modest amounts of filtration. Scale bars represent 200 microns length. All images are collected using identical microscope settings. Images of each organ are contrast enhanced identically.

## Pharmacokinetic and toxicological profiles of the HAV- and LAV-ADCs in normal rats

The pharmacokinetics and toxicity of the HAV- and LAV-ADCs were evaluated in normal rats with escalating single IV doses to assess how the *in vitro* differences in antigen binding affinities of the two ADCs would modulate these parameters in an *in vivo* system. Free payload (MMAE) concentrations were not detected in the blood at any timepoint for either the LAV- or HAV-ADCs following single IV doses of 3-, 6- or 10 mg/kg to rats. No obvious differences between the total mAb and conjugated mAb pharmacokinetic profiles were noted for either ADC following single IV doses of 3-, 6- or 10-mg/kg, indicating the ADCs have comparable and good linker-payload stability in blood across the dose range (Fig 4 and Table 2) consistent with our previous findings for such IgG2-based ADC [25]. A strong dose-dependent effect on blood exposure was confirmed for both ADCs, as reflected in the blood pharmacokinetics curves, which showed a decrease in clearance with increased dose (Fig 4 and Table 2). The non-linear clearance profiles of the two ADCs are largely consistent with a saturable TMDD mechanism as supported by the similar clearance rates at the mid (6 mg/kg) and high (10 mg/kg) dose levels when evaluated within each compound (Table 2). Dose normalized exposure comparisons across the two ADCs robustly show the LAV-ADC has ~2-fold slower blood clearance than the HAV-ADC at equivalent doses (Table 2). The more rapid peripheral clearance of the HAV-ADC in rats is consistent with its higher MET binding affinity measured *in vitro* (Table 1).

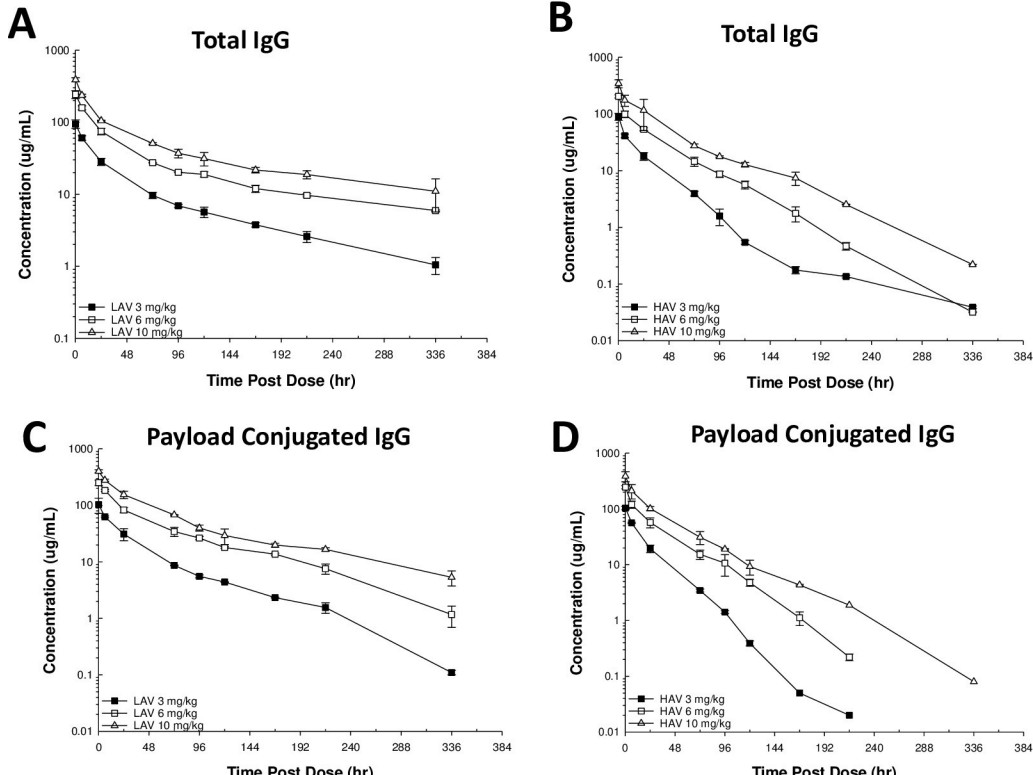

**Fig 4. Pharmacokinetic profiles in of the LAV and HAV ADCs in normal Sprague Dawley rats.** Data are the mean (+/-SD) concentrations for three animals/timepoint for both molecules. The total human IgG (A and B) and payload conjugated IgG (C and D) kinetics of the ADCs were characterized following a single 3-, 6- or 10-mg/kg IV administration of each molecule.

**Table 2. Normal rat mean (± SD) pharmacokinetic parameters for the HAV- and LAV-ADCs.**

| ADC Molecule Measured*# | Dose^ (mg/kg) | $C_{max}$/Dose (kg*µg·mL$^{-1}$·mg$^{-1}$) | $AUC_{inf}$/Dose (hr*kg*µg·mL$^{-1}$·mg$^{-1}$) | CL (mL$^{-1}$·hr$^{-1}$·kg$^{-1}$) | $V_{ss}$ (mL$^{-1}$·kg$^{-1}$) | $T_{1/2}$ (hr) |
|---|---|---|---|---|---|---|
| HAV Total mAb | 3 | 29.9 (2.3) | 483 (11) | 2.1 (0.05) | 60.3 (5.5) | 29.2 (3.6) |
| | 6 | 34.0 (1.9) | 720 (96) | 1.4 (0.2) | 51.5 (2.6) | 33.3 (5.9) |
| | 10 | 34.6 (3.4) | 877 (106) | 1.1 (0.1) | 50.9 (3.8) | 74.8 (2.7) |
| HAV Conjugated mAb | 3 | 34.6 (3.5) | 541 (67) | 1.8 (0.22) | 41.2 (4.6) | 30.6 (4.2) |
| | 6 | 40.9 (1.2) | 794 (25) | 1.3 (0.04) | 41.7 (1.7) | 37.9 (6.8) |
| | 10 | 38.7 (2.6) | 876 (31) | 1.1 (0.04) | 44.2 (1.0) | 80.3 (7.8) |
| LAV Total mAb | 3 | 31.6 (4.8) | 1040 (157) | 1.0 (0.2) | 80.1 (2.4) | 91.0 (6.4) |
| | 6 | 40.5 (2.1) | 1682 (303) | 0.6 (0.1) | 88.8 (3.1) | 167.3 (15.2) |
| | 10 | 38.8 (1.5) | 1685 (224) | 0.6 (0.08) | 97.6 (1.1) | 168.4 (6.4) |
| LAV Conjugated mAb | 3 | 34.4 (1.7) | 919 (120) | 1.1 (0.1) | 53.3 (2.1) | 83.5 (7.9) |
| | 6 | 42 (2.3) | 1541 (132) | 0.6 (0.05) | 42.9 (3.3) | 189.2 (35.6) |
| | 10 | 40.2 (1.9) | 1689 (245) | 0.6 (0.09) | 50.3 (1.0) | 198.4 (84.4) |

*Measure refers to Tab (total antibody) or Conjugated mAb assay used to determine the concentration of the ADC over time.

$C_{max}$/Dose, dose-normalized maximal observed serum concentration; $AUC_{inf}$/Dose, dose-normalized area under the serum concentration curve from time zero extrapolated to infinite time; CL, clearance following IV administration; $V_{ss}$, volume of distribution at steady state; $T_{1/2}$, elimination half-life; ^N = 3 rats/time point. All PK parameters were determined from non-compartmental pharmacokinetic analyses.

#Free payload PK was not reported as free MMAE was undetectable in blood following a single IV administration at any of the dose levels evaluated.

The reported high expression of MET in liver along with the IVM data indicate the mAbs can accumulate within hepatic sinusoidal endothelial cells, leading to strong concerns of increased payload delivery to liver tissue via the target engagement properties of the ADC; as a result, ALT and AST were evaluated as reasonable surrogates of toxicity in liver. Unlike the exposure data, a strong dose-dependent effect on ALT and AST was only observed for the HAV-ADC (Table 3). At doses of 3-, 6- and 10-mg/kg AST increases were 54% (±11%), 83% (±5%) and 234% (±22%) higher than baseline levels, respectively, at peak timepoints for the HAV-ADC. In contrast, the LAV-ADC did not show any overt or dose-dependent AST increases at any of the three dose levels relative to baseline (Table 3) despite its ~2-fold higher peripheral exposure compared to the HAV-ADC. ALT elevations were apparent for the HAV-ADC at 10 mg/kg (~167 ± 39% increase) but no obvious changes in ALT were noted for the LAV-ADC consistent with the rank order in ADC toxicity observed with the AST data (Table 3).

## LAV- and HAV-ADC tumor regression activity in HT29 tumor bearing xenograft rats

The tumor regression activity of the HAV- and LAV-ADCs and isotype control ADC were evaluated in rats implanted with HT29 derived xenografts with escalating single IV doses of 3-, 6- and 10-mg/kg to assess how the *in vitro* differences in antigen binding affinities of the two ADCs affects the HT29 tumor regression pharmacodynamics in an *in vivo* system with TMDD. Despite its ~40-fold lower MET binding affinity, the LAV-ADC showed comparable rates and extents of tumor regression activity as the HAV-ADC at equivalent doses (Fig 5). The therapeutic index (TI = toxic dose ÷ efficacious dose) for the HAV- and LAV-ADCs were estimated using their pharmacodynamic and toxicity data as approximately < 0.3 (TI HAV-ADC = 3 mg/kg ÷ 10 mg/kg) and >1 (TI LAV-ADC = >10 mg/kg ÷ 10 mg/kg), respectively.

**Table 3. Rat mean (± SD) AST and ALT values for the HAV- and LAV-ADCs over time.**

| | | AST (U/L) | | | | |
|---|---|---|---|---|---|---|
| Molecule | Dose^ (mg/kg) | Predose | 72 h | 168 hr | 336 hr | Maximal AST Increase From Predose (%)* |
| HAV | 3 | 100 (4) | 153 (6) | 81 (4) | 79 (3) | 54 (11) |
| | 6 | 94 (3) | 171 (1) | 74 (3) | 65 (3) | 83 (5) |
| | 10 | 98 (11) | 327 (16) | 102 (32) | 94 (2) | 234 (22) |
| LAV | 3 | 86 (3) | 86 (3) | 89 (2) | 79 (12) | 3 (1) |
| | 6 | 79 (9) | 83 (10) | 83 (11) | 80 (6) | 6 (2) |
| | 10 | 82 (4) | 82 (4) | 86 (5) | 82 (2) | 4 (1) |

| | | ALT (U/L) | | | | |
|---|---|---|---|---|---|---|
| Molecule | Dose (mg/kg) | Predose | 72 h | 168 hr | 336 hr | Maximal ALT Increase From Predose (%)* |
| HAV | 3 | 66 (4) | 49 (8) | 78 (18) | 71 (14) | No increase observed |
| | 6 | 66 (7) | 66 (4) | 64 (2) | 57 (2) | No increase observed |
| | 10 | 69 (0) | 184 (27) | 72 (5) | 87 (2) | 167 (39) |
| LAV | 3 | 64 (5) | 63 (14) | 69 (2) | 69 (9) | 8 (9) |
| | 6 | 66 (3) | 66 (2) | 68 (5) | 58 (12) | 2 (3) |
| | 10 | 66 (2) | 67 (2) | 66 (4) | 67 (3) | 2 (0.05) |

^N = 3 rats/time point.

*Reported for the timepoint that peak increase was observed.

## Immunofluorescence detection of MET and the two MET-ADCs in HT29 tumors

Immunofluorescence (IF) staining for MET and the two MET-ADCs in HT29 tumors extracted from the xenograft bearing animals within the aforementioned rat tumor regression studies was utilized as a means to assess the tumor distribution of the molecules. IF was conducted on tumors derived HT29 xenograft rats administered a single IV dose 10 mg/kg from each of the HAV, LAV, and isotype control ADC groups at 24 hours post dose. IF staining of

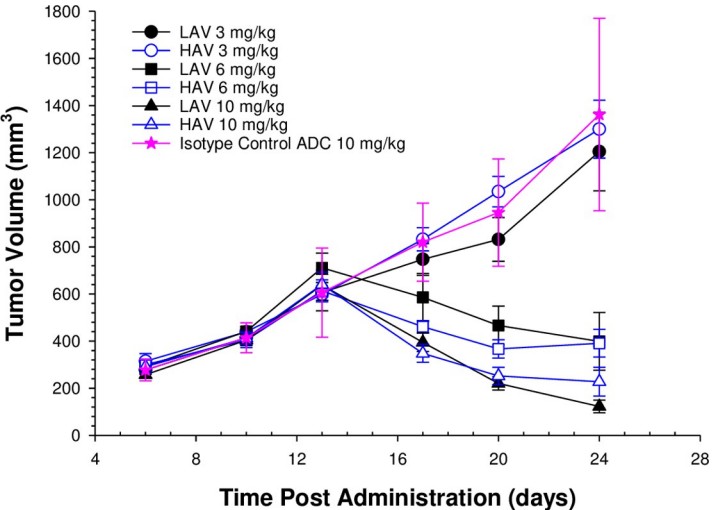

**Fig 5. Tumor regression profiles of LAV- and HAV-ADCs in HT29 tumor bearing rats.** Data are the mean (+/-SD) concentrations for six animals/timepoint for both molecules following a single 3-, 6- or 10-mg/kg IV administration of each molecule. The assessment of the tumor regression activity of an isotype control ADC was included following a single 10 mg/kg IV administration.

tumor tissue for MET and the mAb component of HAV- and LAV-ADCs qualitatively show similar patterns of distribution within the tumor supporting colocalization of MET and the ADCs via target binding, unlike the isotype control ADC which showed no evidence of uptake into the tumors under the same IF staining conditions (Fig 6).

## Biodistribution of radiolabeled HAV- and LAV-ADCs in HT-29 tumor bearing rats

Radiolabeling followed by SPECT/CT imaging of the LAV- and HAV-ADCs was utilized to determine the normal tissue and tumor biodistribution, accumulation and elimination kinetics as a means to dissect the mechanistic basis of the differential therapeutic index findings (ie. both tumor efficacy and toxicity signals). Both non-tissue residualizing (radiohalogen $^{124}$I) and tissue residualizing (radiometal-chelate $^{111}$In-DTPA) labels were leveraged. For ADCs labeled with the radiohalogen $^{124}$I, following antigen mediated tumor or tissue (ie. TMDD) internalization and intracellular degradation, the resultant catabolic products are cleared from the cell; thus, this labeling approach provides insight into ADC tissue/tumor elimination kinetics. For ADCs labeled with the radiometal–chelate $^{111}$In-DTPA, the resultant catabolized radiometal–chelate (typically an amino acid adduct) are trapped within the tissue/tumor cells due to the highly polar nature of the chelate; this labeling approach offers information into ADC tissue/tumor accumulation. The different cellular retentions of their respective catabolic products provide complementary information which can be used to understand the rate and extent of ADC tissue/tumor kinetics (radiohalogen) and cumulative tissue/tumor exposure (radiometal–chelate).

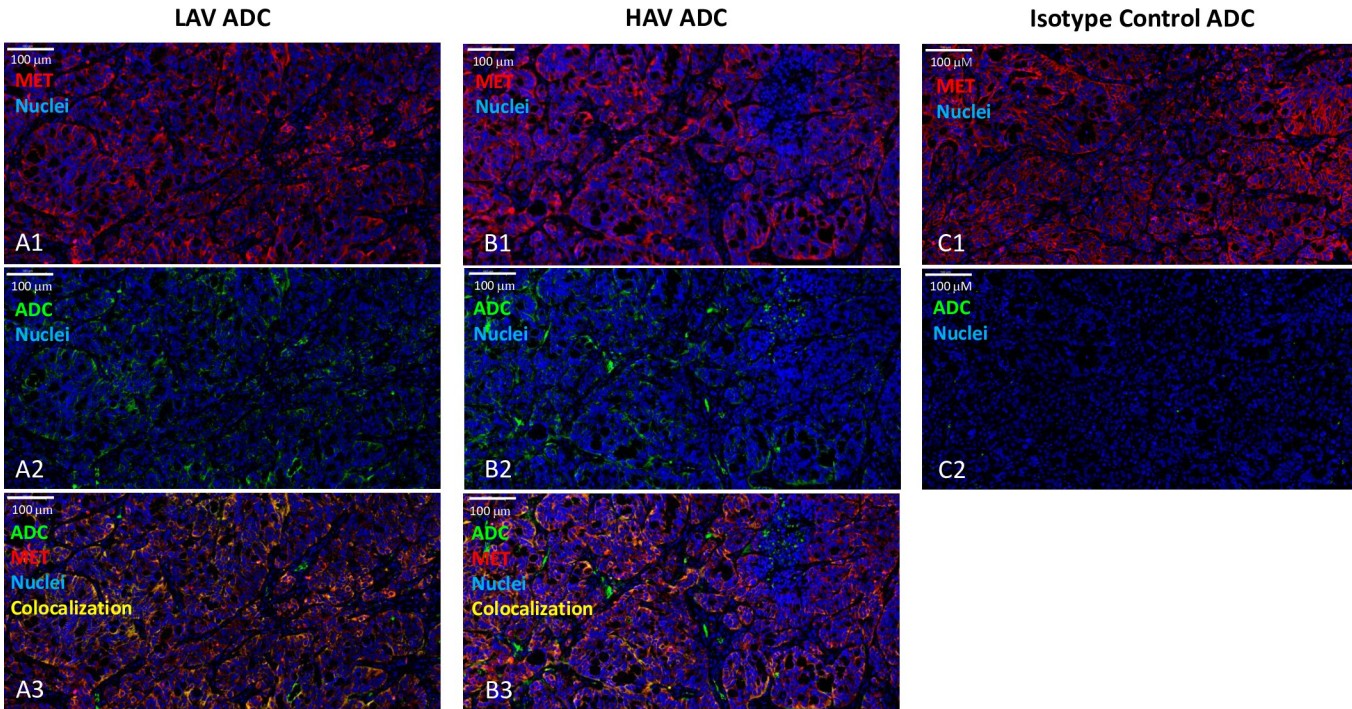

**Fig 6. Representative immunofluorescence detection images.** LAV-ADC (A1, A2 and A3), HAV-ADC (B1, B2 and B3) and isotype control ADC (C1 and C2) in HT29 tumors following a single 10 mg/kg IV administration of each molecule 24 hours post dose. Nuclei were stained with DAPI. Data are shown from a representative HT-29 tumor section from a single rat for each compound. Colocalization shows the overlap of the ADC and MET target. The scale bars represent 100 microns.

Following a single IV injection of [124]I-labeled or [111]In-DTPA labeled MET-ADCs, the tumor and two major highly vascularized MET expressing organs/tissue (liver and kidney) concentrations were measured for each molecule over the course of 168 hours post administration using quantitative non-invasive SPECT/CT imaging (Fig 7). Quantitative non-invasive SPECT/CT imaging facilitates longitudinal imaging of a single animal over consecutive days, so that the kinetics of tumor and tissue uptake and elimination may be visualized and quantified. The [124]I data show the LAV-ADC has a slower rate of clearance in blood and MET expressing tissues, including liver, kidney and the HT29 tumor compared to the HAV-ADC (Fig 8A1–8D1). The slower rate of clearance (or higher exposure) is mechanistically consistent with the lower MET binding affinity of the LAV-ADC that leads to reduced MET-mediated TMDD. The [111]In-DTPA findings demonstrate that the HAV-ADC has increased accumulation in normal tissues expressing MET (liver and kidney) but not in the HT29 tumor (Fig 8A2–8D2). The similar extent of accumulation of the LAV- and HAV-ADCs in the HT29 tumor suggests that the ~40-fold lower MET binding affinity of the LAV-ADC did not negatively impact tumor internalization of the LAV-ADC.

### Free payload exposure in liver and tumor tissues following ADC administration to HT29 xenograft rats

While the radiolabeled ADC biodistribution studies provided quantitative insight into tissue and concentrations of the mAb portions of the ADCs, the amount of free MMAE (ie. payload) in tissues could not be determined from these studies. Given the aforementioned ~3-fold higher TI noted for the LAV-ADC compared to the HAV-ADC, we evaluated the free MMAE concentrations in liver and tumor tissues from HT29 xenograft rats as a means to further disseminate the mechanistic basis of the TI observations.

Following a single 6 mg/kg IV injection of LAV, HAV, and the isotype control ADCs to HT29 xenograft rats, pharmacokinetics of free MMAE were measured for each molecule from

**A1.** [124]I-Labeled LAV MET ADC

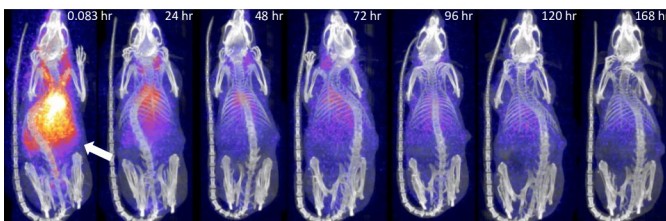

**B1.** [124]I-Labeled HAV MET ADC

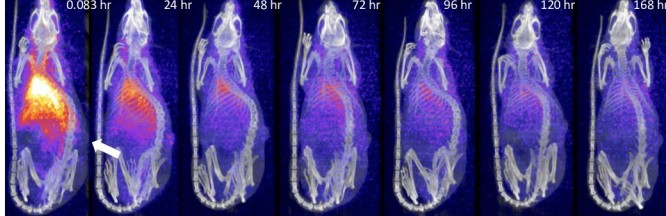

**A2.** [111]In-DTPA-Labeled LAV MET ADC

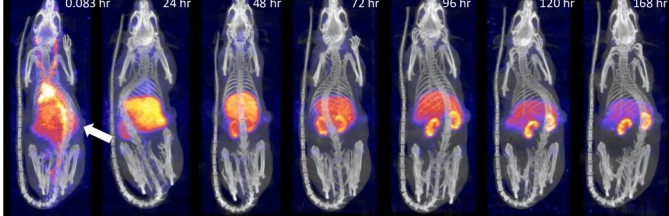

**B2.** [111]In-DTPA-Labeled HAV Affinity MET ADC

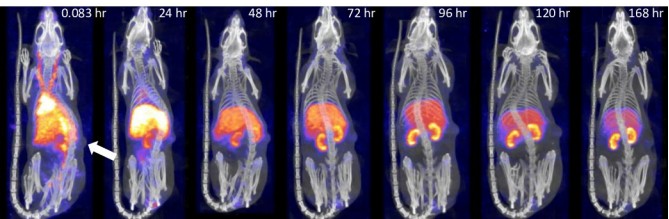

**Fig 7. Representative SPECT/CT images from a single animal.** LAV (A1 and A2) and HAV (B1 and B2) [124]I- (A1 and B1) or [111]In-DTPA- (A2 and B2) labeled MET-ADCs following a single IV administration of ~6 mg/kg (~10 mCi/animal) in male HT29-tumor bearing rats. Images scaled to "fixed percent injected dose (% ID)" are scaled so that the maximum voxel value in the image is equal to a fixed percentage of the dose injected into the animal in mCi (decay corrected to the time of imaging). Voxel values between 0 and the fixed max are scaled linearly. Images are thus quantitatively comparable across animals, because similar color values are representative of similar radioactivity levels. Arrows show the location of the HT29 tumor xenografts on the flank of rats.

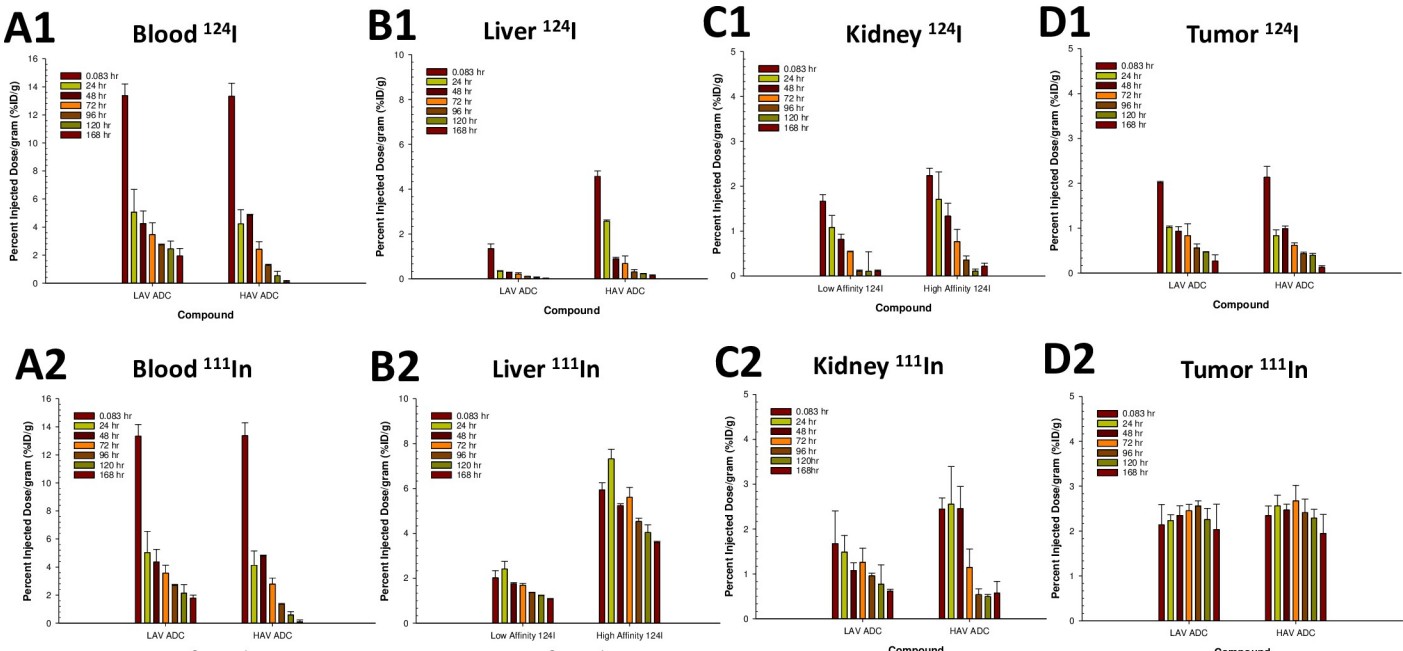

**Fig 8. Mean (+/- SD) concentrations (%ID/g) of the radiolabeled LAV and HAV MET-ADCs following a single IV administration of ~6 mg/kg (~10 mCi/ animal) in male HT29-tumor bearing rats.** [124]I-labeled LAV and HAV MET-ADCs in (A1) blood, (B1) liver, (C1) kidneys and (D1) tumor or [111]In-DTPA-labeled LAV and HAV MET-ADCs in (A2) blood, (B2) liver, (C2) kidneys and (D2) tumor. Data are N = 3/timepoint.

liver and tumor over the course of 168 hours post administration using LC/MS (**S2 Fig**). The isotype control ADC showed ~2-fold lower free MMAE AUC in both liver and tumor relative to the LAV- and HAV-ADCs (Table 4). The lower free payload concentrations for the isotype control ADC compared to the two MET binding ADCs are consistent with the isotype control ADC's lack of TMDD in the liver and tumor tissue. The AUC differences between the pharmacokinetic profiles for the two MET ADCs in the tumor were subtle, with the LAV-ADC showing ~20% higher free MMAE exposure relative to the HAV-ADC (Table 4). The similar levels of free MMAE in the tumors are consistent with observations from the radiolabeled studies which showed comparable accumulation of the LAV- and HAV-ADCs in the HT29 tumor. Within the liver, the two MET-ADCs showed somewhat greater apparent differences in free payload concentrations with the HAV-ADC having ~40% higher free MMAE AUC compared

**Table 4. HT29 tumor bearing rat mean free payload liver and tumor pharmacokinetic parameters for the isotype control, HAV- and LAV-ADCs.**

| ADC Molecule | Tissue | $T_{max}$ (hours) | $C_{max}$ (pg/g) | $AUC_{last}$ (hr*pg/g) |
|---|---|---|---|---|
| **HAV Free MMAE** | Liver | 6 | 16,001 | 758,028 |
| | Tumor | 48 | 28,595 | 2,965,540 |
| **LAV Free MMAE** | Liver | 1 | 17,379 | 557,274 |
| | Tumor | 48 | 31,716 | 3,695,972 |
| **Isotype Control Free MMAE** | Liver | 1 | 4,095 | 307,501 |
| | Tumor | 96 | 11,713 | 1,555,023 |

$T_{max}$, time of maximal observed liver or tumor concentration $C_{max}$, maximal observed liver or tumor concentration; $AUC_{last}$, area under the liver or tumor concentration curve from the first to the last experimental timepoints to the last timepoint. Non-serial sample with N = 3 rats/time point was leveraged as tumor and liver collections involved terminal sample collection. All PK parameters were determined from non-compartmental pharmacokinetic analyses.

to the LAV-ADC in liver (Table 4). The higher concentrations of free MMAE in liver for the HAV-ADC are mechanistically aligned with its increased toxicity relative to the LAV-ADC.

## Discussion

The development of novel ADC therapies represents a promising strategy in the treatment of various cancers [4, 5]. However, expression of targets in normal tissue leads to TMDD and on-target/off-tumor toxicity which is challenging for the clinical development of ADCs. As such, we tested the hypothesis that lowering target affinity of a MET ADC (LAV-ADC) reduces specific binding in normal tissue sinks while retaining sufficient tumor uptake relative to a higher affinity ADC (HAV-ADC). This strategy's framework relies upon differential TMDD biodistribution profiles between the LAV and HAV antibodies in normal and tumor tissues leading to differences in toxicity profiles without impacting the efficacy (illustrated in Fig 1). More specifically, lowering the target affinity was speculated to reduce LAV TMDD in normal MET-expressing tissue (such as liver and kidney) while having comparable concentrations within HT29 tumors relative to the HAV-ADC. It was hypothesized that the higher level of MET expression in HT29 tumor cells would facilitate LAV uptake through an 'avidity-like' effect, which would compensate for the reduced mAb affinity, while sparing TMDD via the majority (or enough) of the antigens in normal tissue sinks and thereby increase the therapeutic index.

Our results validated the aforementioned hypotheses by demonstrating comparable efficacy of LAV- and HAV-ADC in the HT29 xerograft model, elevated liver toxicity observed only with the HAV-ADC, and similar tumor uptake of [111]In-DTPA radiolabeled LAV- and HAV-ADC in HT29 tumors following a 6 mg/kg IV administration (~2.5% ID/g [111]In-DTPA labeled at 24 h; Fig 8). Likewise, comparable HT29 tumor elimination kinetics of [124]I radiolabeled LAV- and HAV-ADCs after the 6 mg/kg IV administration (~1% ID/g [124]I labeled at 24 h; Fig 8) indicate once taken up into the tumor the two molecules have similar rates of catabolism that lead to MMAE release (Table 4). These similar rates of catabolism are further supported by the comparable amounts of free MMAE measured within tumors from animals administered HAV- and LAV-ADC over time (Table 4); although we cannot fully preclude the serendipitous possibility that some free payload in the LAV and HAV tumors may be derived from non-tumor metabolism (ie. other tissues and/or from blood, as well as, the 'bystander' effect), the remarkable similarity in the free payload kinetic profile is more consistent with an antigen-mediated uptake and catabolism mechanism via MET. While tumor concentrations were not evaluated at the higher (10 mg/kg) and lower (3 mg/kg) doses in the radiolabeled studies, the overlapping inhibition curves from HT29 xenograft tumor dose escalation regression profiles indicate the LAV- and HAV-ADCs were likely taken up and eliminated to a similar extent by a MET-mediated mechanism within the tumors (Fig 5). Furthermore, the IF studies on fixed tumor tissue following a 10 mg/kg IV dose confirmed that uptake into HT29 cells was specifically antigen-mediated as there was evidence that both ADCs showed patterns of colocalization with MET unlike the isotype control ADC (Fig 6), supporting antigen-based interactions play a crutial role in the ADC tumor disposition. Combined, the similar extent of accumulation of the MET-ADCs in the HT29 tumor suggests that the lower MET binding affinity of the LAV did not negatively impact its internalization into tumor cells. We speculate that increased avidity as a result of high level MET expression in the tumor results in enhanced affinity for the LAV MET-ADC on tumor cells (Fig 1).

Unlike within tumor, LAV mAb and LAV-ADC clearly have reduced the accumulation and elimination kinetics within normal tissues compared to HAV mAb and HAV-ADC (Figs 7 and 8). The lowered tissue uptake is consistent with LAV-ADC having reduced liver toxicity compared to its higher affinity counterpart at a comparable dose (6 mg/kg). Even following a

single 3 mg/kg IV HAV-ADC dose, the highly elevated transaminases indicate there is differential and increased accumulation of the HAV-ADC in liver tissue, which is consistent with its enhanced toxicity. These observations are in spite of the available ~2-fold higher LAV-ADC concentrations present in the blood (Table 2). At a cellular level, the data suggest the increased MET binding affinity of HAV-ADC leads to greater normal tissue mediated TMDD which facilitates increased release of the payload and elicits the liver toxicity signals (Table 3). Given this, we expected to observe striking increases in liver free payload concentrations for HAV-ADC relative to LAV-ADC, as this would connect with sparing these tissues from unintended payload exposure. Intriguingly, our evaluations showed that the free MMAE levels were only modestly higher in liver tissue for HAV-ADC versus the LAV-ADC (<1.4 fold difference) (Table 4). While this may be linked to a number of experimental factors including incomplete characterization of MMAE metabolites, the permeability of MMAE, limited concentration versus time profile data and the small sample size of the evaluation, it is thought-provoking to speculate there may be some connectivity to cellular disposition. Indeed, the strong accumulation of HAV (but not the LAV) mAb within sinusoidal endothelia following a single 6 mg/kg dose in the IVM studies supports the contention that exceptionally higher liver toxicity may be in part associated with increased specific delivery of MMAE to hepatocytes. As such, these data indicate the ~3-fold therapeutic index differences between the HAV- and LAV-ADCs may be related to several factors including the extent of ADC uptake, the rate of ADC elimination and the cellular disposition payload released in both normal tissues and tumor. Another plausible advantage of a lower affinity ADC, although not specifically investigated here, may be its differential ability to penetrate solid tumor microenvironment as has been broadly studied in microspacial distribution [31]. These tumor cell-specific distribution investigations have suggested that lower affinity mAbs and ADCs can have better tumor penetration (and by extension efficacy) due to the theory of a 'binding site barrier'. Nearly four decades ago Weinstein and colleagues postulated that in tumor cells expressing receptors in very high copy numbers and/or when an antibody binds tumors antigens with very high affinity, it is possible that the molecule would get 'stuck and consumed' by the outermost tumor cells close to the peripheral blood supply and thus have limited spatial penetration throughout the deeper portions distal to the vasculature [32]. Although outside the scope of this work, additional research to characterize payload and ADC tumor penetration would provide insight on these factors.

## Conclusions

Herein, we have demonstrated proof-of-concept that lowering the target affinity of mAb is a simple and elegant strategy to improve the therapeutic window for MET-ADC. With a ~40-fold difference in binding affinity, HAV- and LAV-ADCs showed comparable rates and degrees of tumor regression activity (at equivalent doses); however, the reduced affinity of the LAV-ADC led to an improved liver toxicity profile with no overt changes in liver function tests (AST and ALT). As a result, the estimated therapeutic index for LAV-ADC was at least 3 times greater than the HAV-ADC. Our IVM and radiolabel biodistribution studies revealed the mechanistic basis of the TI improvement, providing evidence that decreasing the affinity of the ADC reduced the TMDD to MET expressed in normal tissues (while maintaining sufficient uptake of the ADC), as well as delivery of the payload to tumor which over-expresses MET, likely due to avidity. These particularly promising findings suggest a potential clinically applicable strategy to increase the ADC therapeutic window utilizing existing linker-payload technologies.

## Supporting information

**S1 Fig. Distribution of DL488 fluorescently labeled HAV and LAV in the liver of living rats.** Multiphoton fluorescence excitation images of the livers of living Munich-Wistar-Fromter rats were collected 2 hours after intravenous injection of 6 mg/kg DL488-HAV (Panel A) or DL488-LAV (Panel B). Indicated regions are reproduced at 3X magnification in insets. Similar to the results obtained at 24 hours, punctate fluorescence is observed in rats injected with DL488-HAV but not DL488-LAV. (C) Time series of images collected from the same region of the liver prior to (left panel) and then again 1 and 2 hours after intravenous injection of DL488-HAV (middle and right panels, respectively). In this study, DL488-HAV accumulation was particularly pronounced, becoming apparent within one hour, and continuing to accumulate over the next hour. Neither construct is detectibly associated with hepatocytes, as identified by their brown autofluorescence, arranged in linear "cords". Bright white punctate fluorescence reflects broad-spectrum autofluorescence of vitamin A in stellate cells. Blue signal derives from second harmonic generation, likely from collagen. Scale bars represent 200 microns in panels A and B, and 50 microns in panel C. All images are collected using identical microscope settings and contrast enhanced identically.
(TIF)

**S2 Fig. Free MAAE in tumor and liver following ADC administration to HT29 tumor bearing rats.** Pharmacokinetic profiles of free MMAE (payload) following a single 6 mg/kg IV administration of the isotype control ADC or LAV and HAV ADCs within liver and tumor tissues in HT29 tumor bearing rats. Data are the mean (+/-SD) concentrations for three animals/timepoint for each of the molecules. The free MMAE exposure following administration of the ADCs were characterized for 168 hours post the single administration. Non-serial sample with N = 3 rats/time point was leveraged as tumor and liver collections involved terminal sample collection.
(TIF)

## Acknowledgments

The authors kindly acknowledge Bernice Ellis, Jason Anderson, Linda Schirtzinger and Chi-Kin Chow for their support of the work, helpful discussions and/or careful review of the manuscript.

## Author Contributions

**Conceptualization:** Amita Datta-Mannan, Ling Liu, Jirong Lu, Kenneth W. Dunn, Yiqing Feng.

**Data curation:** Amita Datta-Mannan, Hiuwan Choi, Zhaoyan Jin, David J. Stokell, Anthony T. Murphy, Kenneth W. Dunn, Michelle M. Martinez.

**Formal analysis:** Amita Datta-Mannan, Zhaoyan Jin, David J. Stokell, Anthony T. Murphy, Kenneth W. Dunn, Michelle M. Martinez.

**Investigation:** Amita Datta-Mannan, Kenneth W. Dunn.

**Methodology:** Amita Datta-Mannan, Hiuwan Choi, David J. Stokell, Anthony T. Murphy, Kenneth W. Dunn.

**Project administration:** Amita Datta-Mannan.

**Supervision:** Amita Datta-Mannan, Kenneth W. Dunn, Yiqing Feng.

**Writing – original draft:** Amita Datta-Mannan, Yiqing Feng.

**Writing – review & editing:** Amita Datta-Mannan, Yiqing Feng.

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
