## [Decision Letter · Decision Letter 0]

25 Jul 2023

PONE-D-23-19969Reducing target binding affinity improves the therapeutic index of anti-MET antibody-drug conjugate in tumor bearing animalsPLOS ONE

Dear Dr.
Datta-Mannan,

Thank you for submitting your manuscript to PLOS ONE. After careful consideration, we feel that it has merit but does not fully meet PLOS ONE’s publication criteria as it currently stands. Therefore, we invite you to submit a revised version of the manuscript that addresses the points raised during the review process.

We look forward to receiving your revised manuscript.

Kind regards,

Satish Rojekar, Ph.D.

Academic Editor

PLOS ONE

Journal Requirements:

6. Please amend the manuscript submission data (via Edit Submission) to include author Hiuwan Choi, Zhaoyan Jin, Ling Liu, Jirong Lu, David J. Stokell, Anthony T. Murphy, Kenneth W. Dunn, Michelle M. Martinez and Yiqing Feng.

7. We note that you have included the phrase “data not shown” in your manuscript. Unfortunately, this does not meet our data sharing requirements. PLOS does not permit references to inaccessible data. We require that authors provide all relevant data within the paper, Supporting Information files, or in an acceptable, public repository. Please add a citation to support this phrase or upload the data that corresponds with these findings to a stable repository (such as Figshare or Dryad) and provide and URLs, DOIs, or accession numbers that may be used to access these data. Or, if the data are not a core part of the research being presented in your study, we ask that you remove the phrase that refers to these data.

8. Please include your full ethics statement in the ‘Methods’ section of your manuscript file. In your statement, please include the full name of the IRB or ethics committee who approved or waived your study, as well as whether or not you obtained informed written or verbal consent. If consent was waived for your study, please include this information in your statement as well. 

Reviewers' comments:

Reviewer's Responses to Questions

**Comments to the Author**

1. Is the manuscript technically sound, and do the data support the conclusions?

Reviewer #1: Yes

Reviewer #2: Yes

2. Has the statistical analysis been performed appropriately and rigorously? 

Reviewer #1: Yes

Reviewer #2: Yes

3. Have the authors made all data underlying the findings in their manuscript fully available?

Reviewer #1: Yes

Reviewer #2: Yes

4. Is the manuscript presented in an intelligible fashion and written in standard English?

Reviewer #1: Yes

Reviewer #2: Yes

5. Review Comments to the Author

Reviewer #1: The current research manuscript by Datta-Mannan et al. “Reducing target binding affinity improves the therapeutic index of anti-MET antibody-drug conjugate in tumor bearing animals” showed immense potential and outstanding match for publishing in the PLOS One Journal. I found the research is excellent and highly recommend it for publication without any revisions. Kindly re-check and correct the formatting error in the manuscripts if required.

Reviewer #2: The authors have postulated a hypothesis and supported the hypothesis using well designed experiments. However, even though this is a proof of concept study for high affinity and low affinity antibodies for delivery of ADC , I would like to point out that almost similar research has been previously reported (https://www.nature.com/articles/s41586-022-05673-2 )

I would like to request clarification on certain points

1.) Administration of antibody without the cytotoxic payload prior to ADC administration has been proven to be beneficial therapeutically. What advantage does LAV present over the pre administration of plain antibody?

2.) Based on this information and considering the longer t1/2 of LAV, it would be pertinent to have data evaluating the effect of pre-administration of unlabeled HAV on the therapeutic efficacy of LAV ADC.

3.) Although figure 6 present co-localization data, confocal based co-localization study in the intracellular endosomal area will serve as a direct evidence of ADC uptake. Additionally, some explanation regarding bystander effect or a mechanistic study to determine if MMAE from LAV is providing bystander effect would be beneficial

4.) How does the lower affinity by spr translate to number of MaB binding per cell over expressing MET?

5.) While the authors provided liver and kidneys based data, it was interesting to understand the exclusion of the lung,a highly perfused organ having high occurrence of MET positive cancers, from the study especially using the IVT microscopy.

6. PLOS authors have the option to publish the peer review history of their article (what does this mean?). If published, this will include your full peer review and any attached files.

Reviewer #1: **Yes: **Dr. Saurabh Sharma

Reviewer #2: No

<quillbot-extension-portal></quillbot-extension-portal>

---

## [Author Response · Author response to Decision Letter 0]

11 Oct 2023

Journal Requirements:

COMPLETED

2. Thank you for your submission to PLOS ONE. We note that your study design may include death of a regulated animal as a likely outcome or planned experimental endpoint. At this time, we request that you please report additional details in your Methods section regarding animal care and use for the survival study, as per our editorial guidelines 

COMPLETED - included marked and unmarked revised manuscripts

3. For easy reference, we have attached a checklist that may be relevant for your submission. Please complete all items on the checklist at the following link: COMPLETED and included in revised submission

4. We note your current Data Availability statement: "Some data (including, but not limited to mAb sequences and proprietary expression) cannot be shared publicly without Eli Lilly & Company legal authorization. All data was generated by Third Party Sources thus we are unable to provide DOIs."

Please describe the methods of data collection and clarify whether the authors of the present study had any special access privileges in accessing the data which other interested researchers would not have. 

There is no special access privileges for the authors

COMPLETED - All data has been made available in the manuscript and supplemental files.

---

## [Editor Report · Decision Letter 1]

18 Oct 2023

Reducing target binding affinity improves the therapeutic index of anti-MET antibody-drug conjugate in tumor bearing animals

PONE-D-23-19969R1

Dear Dr. Amita Datta-Mannan,

We’re pleased to inform you that your manuscript has been judged scientifically suitable for publication and will be formally accepted for publication once it meets all outstanding technical requirements.

Kind regards,

Satish Rojekar, Ph.D.

Academic Editor

PLOS ONE

Additional Editor Comments (optional):

Dear Dr. Amita Datta-Mannan,

I am delighted to inform you that the revised manuscript has been significantly improved in quality and clarity, making it a strong candidate for acceptance. Based on my assessment and reviewer comments, I recommend accepting the manuscript in its current form. Therefore, I am pleased to announce that the final decision for the manuscript is "Accepted."

Best regards,

Dr. Satish Rojekar

PLOS ONE
---

## [Editor Report · Acceptance letter]

3 Apr 2024

PONE-D-23-19969R1 

PLOS ONE

Dear Dr. Datta-Mannan, 

I'm pleased to inform you that your manuscript has been deemed suitable for publication in PLOS ONE. Congratulations! Your manuscript is now being handed over to our production team.

Kind regards, 

on behalf of

Dr. Satish Rojekar 

Academic Editor

PLOS ONE